# Finger Kinematics during Human Hand Grip and Release

**DOI:** 10.3390/biomimetics8020244

**Published:** 2023-06-08

**Authors:** Xiaodong Li, Rongwei Wen, Dehao Duanmu, Wei Huang, Kinto Wan, Yong Hu

**Affiliations:** 1Shenzhen Institute of Research and Innovation, The University of Hong Kong, Shenzhen 518057, China; lixd@hku-szh.org (X.L.);; 2Orthopedics Center, The University of Hong Kong-Shenzhen Hospital, Shenzhen 518053, China; 3Department of Rehabilitation, The Second Affiliated Hospital of Guangzhou Medical University, Zhanjiang 524002, China; 4Department of Orthopaedics and Traumatology, The University of Hong Kong, Hong Kong, China

**Keywords:** finger kinematics, motion coordination, grasping and release, robotic hand, bio-inspired

## Abstract

A bionic robotic hand can perform many movements similar to a human hand. However, there is still a significant gap in manipulation between robot and human hands. It is necessary to understand the finger kinematics and motion patterns of human hands to improve the performance of robotic hands. This study aimed to comprehensively investigate normal hand motion patterns by evaluating the kinematics of hand grip and release in healthy individuals. The data corresponding to rapid grip and release were collected from the dominant hands of 22 healthy people by sensory glove. The kinematics of 14 finger joints were analyzed, including the dynamic range of motion (ROM), peak velocity, joint sequence and finger sequence. The results show that the proximal interphalangeal (PIP) joint had a larger dynamic ROM than metacarpophalangeal (MCP) and distal interphalangeal (DIP) joints. Additionally, the PIP joint had the highest peak velocity, both in flexion and extension. For joint sequence, the PIP joint moved prior to the DIP or MCP joints during flexion, while extension started in DIP or MCP joints, followed by the PIP joint. Regarding the finger sequence, the thumb started to move before the four fingers, and stopped moving after the fingers during both grip and release. This study explored the normal motion patterns in hand grip and release, which provided a kinematic reference for the design of robotic hands and thus contributes to its development.

## 1. Introduction

The hand is a crucial component of the human body, and it can handle numerous precise and complex tasks. The versatility of the hand benefits from its sophisticated anatomical features, as it comprises many bones and is driven by numerous muscles [1]. Additionally, the hand is controlled by a complicated neural system that configures the fingers in a suitable way to exert fine movements on different objects [2]. Inspired by human hands, researchers are committed to producing robotic hands by designing mechanical structures and developing control strategies, and have obtained many impressive achievements [3]. Among them, the Okada hand [4] and Utah/MIT hand [5] are usually considered among the representatives of early dexterous robotic hands. An important development trend of robotic hands is anthropomorphic robotic hands, which mimic the biological characteristics of human hands, to achieve powerful manipulations close to human hands. In recent years, a number of anthropomorphic robotic hands have been developed [6,7]. Most of these robotic hands adopt a rigid body that mimics the anatomical structure of a human hand, and a transmission method based on tendon-driven mechanisms or linkage-driven mechanisms. Similarly, many soft robotic hands have been developed whose bodies are typically made of soft material [8,9]. Soft robotic hands are close to human hands in appearance, but they face challenges in imitating human hand grasping. In general, although the robotic hand research has made great progress in recent decades, there is still a big gap between robot and human hands in terms of dexterity and compliance. Endowing the robotic hands with sufficient biological characteristics to improve grasping functionality are the core issue in bionic robotic hand research. The analysis of human finger movement reveals the control and motion patterns of hands, which helps to optimize the design and actuation of robotic hands [10]. Therefore, understanding the kinematics of human fingers is essential to improve the grasping performance of robotic hands.

Hand gripping or grasping is a basic movement that involves flexion and extension of multiple joints. Up to now, the studies on human hand grip mainly focus on the range of motion (ROM), including active, passive and functional ROMs [11,12,13,14,15], which are measured statically. Many robotic hands claimed that their ROMs can reach or be close to that of normal hands [16,17]. As hand grip and release is a dynamic process, a dynamic ROM would be meaningful, and it may differ from the ROMs measured statically. Based on our knowledge of the literature, dynamic ROM during hand grip and release has not been well-addressed. Thus, a dynamic ROM of hand joints is an important index to be investigated. In addition, motion coordination is a critical characteristic of human hands, which should be a priority in the design of robotic hands. Some robotic hands tried to reproduce the human finger motion trajectory as much as possible by optimizing the structural design and control strategies [18,19,20]. However, it is difficult to replicate the trajectory of human hands on a robotic system [21]. Additionally, individual differences in finger motion trajectories are large. A practical solution may be to develop robots that conform to motion coordination. Motion coordination is mainly reflected in the temporal relationship among the joints during hand motion. Several studies have investigated joint motion coordination during finger flexion and extension [22,23,24,25,26]. However, there is a problem in these studies, which lies in their conflicting observations, leading to the controversy. Furthermore, the finger sequence was less considered in previous analyses of finger movement. The motion coordination during finger flexion and extension has not been fully characterized. Therefore, comprehensive examination of the joint and finger sequences is a necessity for understanding hand kinematics.

The purpose of this study was to determine the finger kinematics during hand grip and release, in order to figure out normal hand motion patterns. Specifically, dynamic ROM and peak velocity of the finger joints were measured. Additionally, the joint sequence and finger sequence during flexion and extension were investigated.

## 2. Materials and Methods

### 2.1. Participants

Twenty-two individuals with no hand abnormalities participated in the study (9 women and 13 men; age 28.32 ± 6.40 years). All participants were right-hand dominant. Before the experiment, all participants had understood the experimental content and provided the written informed consent.

### 2.2. Procedures

Hand motion was detected by an elastic sensory glove with 15 sensors [27]. Specifically, the flexion and extension of 14 joints were recorded, including the interphalangeal (IP) and metacarpophalangeal (MCP) joints of the thumb, and the distal interphalangeal (DIP), proximal interphalangeal (PIP) and MCP joints of four long fingers. One advantage of this glove is that its elasticity expands the range of use by effectively reducing the impact of hand size on the measurement accuracy.

During data collection, the participants sat on a chair and placed the right forearm on an armrest, while keeping the palm faced down. They were asked to use the right hand to perform a grip and release motion as quickly as possible, while trying to ensure the fingers at maximum flexion when the hand is closed and at maximum extension when the hand is open. Rapid finger motion can avoid artificial motion patterns, thus reflecting natural patterns. Specifically, the participants were required to complete 10 cycles of motions within 5 s. Normal people are believed to perform more than 20 cycles in 10 s [28].

### 2.3. Measures

The maximum and minimum angular values of finger joints in each cycle were calculated, as well as the peak velocity of joints during the flexion and extension phases.

On the other hand, the time normalization of data was required for the analysis of motion coordination, because the actual duration of motion varied basically from cycle to cycle. As shown in Figure 1, the data corresponding to the flexion and extension states of fingers were segmented, and then converted into percentages to represent the movement progress. In one cycle, 0~50% is for flexion and 50~100% is for extension. For each participant, the normalized data segments that belonged to each motion state were averaged across multiple cycles. A diagram showing finger flexion and extension and the corresponding normalized angular curves of the three joints from one participant is found in Figure 2. To assess the joint sequence comprehensively, three phases of movement were considered, including the beginning, the middle, and the end. Thus, for each joint in the two types of motion, three motion feature points were identified, including the initiation point, the maximum velocity point, and the completion point of motion. Specifically, a point was regarded as the initiation or completion point of motion if either the relative amplitude change with respect to the starting point or the ending point was greater than or equal to 3%, or the ratio of the relative amplitude change and difference of normalized time between the point and the starting or ending point was greater than or equal to 1. Additionally, the joint sequence at the middle of movement was determined by the position of the maximum velocity of the joints. An example of the determination of feature points of three joint angular curves is shown in Figure 2.

Note that in this study, the initiation and completion points did not always correspond exactly to the start and end of joint motion. For the feature points of finger, the earliest initiation point of two or three joints was considered to be the initiation point of the finger, and the last completion point of any joint was considered to correspond to the completion point of the finger. The determination of feature points of fingers is also shown in Figure 2.

### 2.4. Statistical Analysis

For the kinematic variables related to angle and velocity, the mean value for each participant over 10 cycles was calculated, and then the mean and standard deviation (SD) of all participants were calculated. Additionally, for the same type of joints from different fingers, the one-way analysis of variance (ANOVA) was used to determine any significant differences among the fingers.

In addition, average values of the locations of the feature points for all participants were calculated to determine the joint sequence and finger sequence. The paired *T*-test was performed to compare the locations of motion feature points between the joints within each finger. Similarly, the time differences for the feature points between different fingers were evaluated via the paired *t*-tests.

## 3. Results

### 3.1. Dynamic ROM

The dynamic ROM for the 14 finger joints during hand grip and release motion are shown in Figure 3, of which the specific data are shown in Appendix A in Appendix A. PIP joint of long fingers had the largest ROM, followed by MCP and DIP joints. The results of ANOVA are shown in Table 1, indicating that there was no significant difference in the ROM among four long fingers on PIP joint and MCP joint, except DIP joint. Post hoc test showed that the DIP ROM of the index finger was significantly smaller than that of three ulnar fingers (index finger and middle finger: *p*-value = 0.001; index finger and ring finger: *p*-value = 0.002; index finger and little finger: *p*-value = 0.023). However, there was no significant difference in DIP ROM amongst the middle finger, ring finger and little finger.

### 3.2. Peak Velocity

According to Table 2, ANOVA indicated that peak velocity for each kind of joint did not differ across the four long fingers during flexion. Likewise, there was no significant difference in peak extension velocity of the same joint among long fingers. The mean peak velocities during flexion and extension for three joints are shown in Table 3, and the detailed results of each participant are shown in Appendix A in Appendix A. The PIP joint had the highest peak velocity, followed by MCP and DIP joints, during both flexion and extension. Similarly, peak velocity for the thumb was significantly higher at the IP joint than at the MCP joint.

### 3.3. Joint Sequence

The averaged time relationships amongst the feature points of finger joints at different phases during hand grip and release are shown in Figure 4, of which the specific data are shown in Appendix A in Appendix A. The joint sequences were determined on the basis of feature points. As shown in the sub-figures of Figure 4, the joint sequences were not absolutely fixed among four long fingers. In most cases, the differences in the joint sequences among four fingers lay in the positions of the DIP and MCP joints. Comparing the joint sequences of the same finger at three phases, it is found that the joint sequence may vary with motion progression. However, the impact of the motion process on the joint sequence of a finger was minimal.

Figure 4 appears to indicate that the flexion and extension motions had opposite joint sequences. More exactly, the reverse sequences only appeared on certain pairs of joints. Both the DIP and MCP joints preceded the PIP joint during finger extension, while the PIP joint preceded the DIP and MCP joints during the flexion process in most cases. However, the temporal relationship between the DIP and MCP joints was not fixed.

### 3.4. Finger Sequence

The averaged time relationship on finger feature points amongst five fingers at different phases during hand grip and release are shown in Figure 5, of which the specific data are shown in Appendix A in Appendix A. At the beginning of hand gripping, the thumb was significantly ahead of four long fingers, while the little finger was clearly in the final position. However, the thumb did not stop moving until after four fingers had completed their gripping. The finger sequence during hand release is similar to that during hand gripping.

## 4. Discussion

Hand grip and release are holistic movements that involve all five fingers. It is clear that the joints of a same kind shared similar dynamic ROM. The comparison among the dynamic ROM of DIP, PIP and MCP joints shows that the PIP joint has the largest dynamic ROM. Additionally, it is found that the DIP ROM of the index finger was smaller than the other three long fingers during rapid grip and release. A previous study based on static measurement has reported no significant differences on active DIP ROM amongst four long fingers [29]. When the hand was fisted, the thumb performed the flexion motion, accompanied by abduction. As a result, the index finger probably stopped flexing when its tip reached the thenar eminence, leading to a smaller DIP flexion when clenching the hand into a fist. It indicates that the dynamic ROM of the hand joints are influenced by many factors, making it different from the static ROM.

ANOVA indicated no significant differences in peak flexion velocity or peak extension velocity among all four fingers, demonstrating the consistency of finger motion for holistic movements. For long fingers, the peak velocity was largest at PIP joint, followed by MCP and DIP joints, regardless of whether it was during flexion or extension. Similarly, the PIP joint had a higher peak velocity than MCP joint in fast index finger flexion [30] and in rapid thumb–index finger pinch movement [31]. According to the results, the PIP joint not only has the largest dynamic ROM, but also has the largest peak velocity, fitting with the description that PIP joints account for the majority of the finger grasping capability [32]. The results suggest that a robotic hand needs to pay great attention to the PIP joint to ensure that the actuator of the PIP joint has adequate ability to flex and extend.

The analysis of joint sequences and finger sequences during hand grip and release motion revealed much information of finger motion patterns, which reflect the functionality of the hand. The results showed that, although the joint sequences were not absolutely identical among all fingers, they shared some commonalities. On the other hand, previous studies have implied that the finger joint sequence would change during the motion process [33,34]. However, no study has fully characterized the joint sequences at the beginning, middle and end of the flexion and extension of fingers. It is found in this study that, even if the three-joint sequence was not identical in all the phases, the relationships between some joints were consistent. To sum up, long fingers of the dominant hand shared a consistent pattern during grip and release motions. Specifically, the flexion of long fingers first appeared in the PIP joint, while the extension started in the DIP or MCP joint. Moreover, these joint sequences were independent of the phases of motion process, demonstrating the stability of finger motion coordination.

It is known that the fingers are controlled by several muscles, and most of these muscles act over the phalanges through the tendons, which are inserted into the bones [35]. As for the PIP and MCP joints, the flexor digitorum profundus (FDP) and the flexor digitorum superficialis (FDS) connect them for flexion. Motion generated by the FDP and FDS at the PIP joint was found to occur ahead of the motion at the MCP joint [36]. This may be one reason why the PIP joint moved prior to the MCP joint during flexion in this study. For the DIP and PIP joints, they are generally considered as hinge joints capable only of flexion and extension [37]. A “link ligament” between DIP and PIP joints, known as the oblique retinacular ligament or the Landsmeer ligament, acts as a dynamic tenodesis [38]. It is believed that the separate movement of these two IP joints is nearly impossible [39], and a nearly linear linkage between the DIP and PIP joints was observed during finger flexion and extension [40,41]. Anatomically, both the FDP and extensor digitorum communis (EDC) are connected to the phalanges in relation to the DIP and PIP joints. The cadaveric study showed that the FDP generates motion simultaneously at the DIP and PIP joints when the fingers are bent [36]. However, some researchers believed that the DIP joint movement is likely to lag behind that of the PIP joint because the FDP has a longer moment arm across the DIP joint than the PIP joint [42]. In many studies involving finger flexion, the PIP joint was observed to move prior to the DIP joint [33,34]. Clearly, the coordinated motion of fingers is the result of complex neuro-musculo-skeletal interactions [43]. Therefore, for bionic robotic hands, efforts should be made in terms of mechanical structure and control method to achieve a coordinated movement similar to human hands.

The joint sequence of the thumb is different from that of the long fingers. The MCP-IP sequence occurred most frequently in the thumb flexion in this study, which was also observed during the cylinder grip [34] and during the thumb opposition [44]. Similarly, the probability of the IP-MCP sequence was relatively larger during extension. It showed that the relationship between MCP and IP joints of the thumb is not the same as that of the long fingers, which may confirm that the mechanism of thumb motion is different from that of the long fingers [45]. In general, for all five fingers, the joint sequence during the flexion process is contrary to that in the extension process. For a hand robot, it is better to conduct movements in appropriate joint sequence, which helps improve its grasping performance.

In characterizing the finger sequence during hand gripping and release, there appears to be a time gap in the motion between the thumb and four long fingers. As the participants were asked to place their thumb outside of the fist during fist clenching, it was expected that the thumb would be the last appendage to stop moving at the end of hand gripping. Similarly, the participants had to move their thumb firstly to unlock the remaining fingers at the beginning of hand release. In long fingers, the motion was highly synchronized due to their similar biological structures. This finger sequence supports to some extent the view that the thumb is the most independent finger [46]. Therefore, in terms of triggering the timing of motion, if the hand robot handles the thumb and the four long fingers independently, the executed motion would be closer to natural grip and release.

Undoubtedly, grasping is one of key issues in the study of robotic hands. In order to enable robotic hands to achieve effective grasping, it is recommended to establish a database of human grasps and develop the models on grasps [47]. Learning human hand motion and generalizing control strategies from human demonstration, and mapping human skills to robotic hands, enhance the functionality and reliability of robots [48,49]. Therefore, it is of great significance to collect kinematic information and analyze motion patterns from the hands of healthy people. This study provided quantitative data and qualitative patterns of human fingers during hand grip and release, which can serve as a reference for implementing grasping in robotic hands. However, there were several limitations in this study. This study focused on the most basic hand movement, but different finger motion patterns may be exhibited when hand performs other functional activities. On the other hand, only the dominant hand was involved in this study. Although a previous study has demonstrated significant similarities in grasping between dominant and non-dominant hands [50], some of the results in this study might not be applicable to non-dominant hands. Therefore, in the future work, we plan to explore different types of grasping and the kinematics of non-dominant hands to extract comprehensive human grasping patterns. Furthermore, the motion patterns of patients with hand dysfunction will be studied, in order to determine the difference between normal patterns and impaired patterns, which is particularly crucial for hand rehabilitation devices.

## 5. Conclusions

Finger kinematics during human hand grip and release was investigated in this study. Segmental findings on motion patterns showed good correspondence with the literature. Furthermore, more kinematic features were involved in this study, as the dynamic ROM, joint sequence and finger sequence were described. This study presented hand motion features from healthy people, which can serve as the basic information for designing bionic robotic hands. For example, an exoskeleton that aims to enhance hand function of healthy people needs to have the same motion pattern as healthy human hand. Meanwhile, the findings in this study can be used as a baseline for investigating the motion patterns of people with hand injuries and disabilities, and support the design of hand rehabilitation robots. Overall, analysis of the kinematic variables helps to reveal the nature of the finger movement, and is of great reference value for the development of the robotic hand. Bionic robotic hands with normal motion patterns are believed to have the functionality closer to human hands.

## Figures and Tables

**Figure 1 biomimetics-08-00244-f001:**
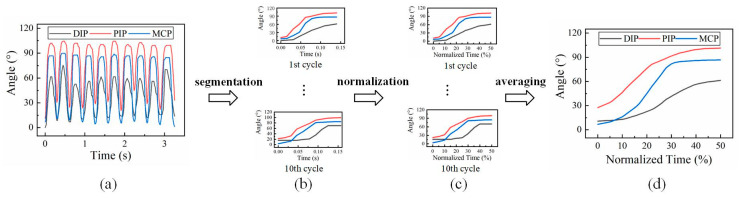
The process of time normalization: (**a**) original data of index finger from a participant, (**b**) 10 cycles of flexion motion segmented from original data, (**c**) normalized data for 10 cycles, and (**d**) averaged data for flexion over 10 cycles.

**Figure 2 biomimetics-08-00244-f002:**
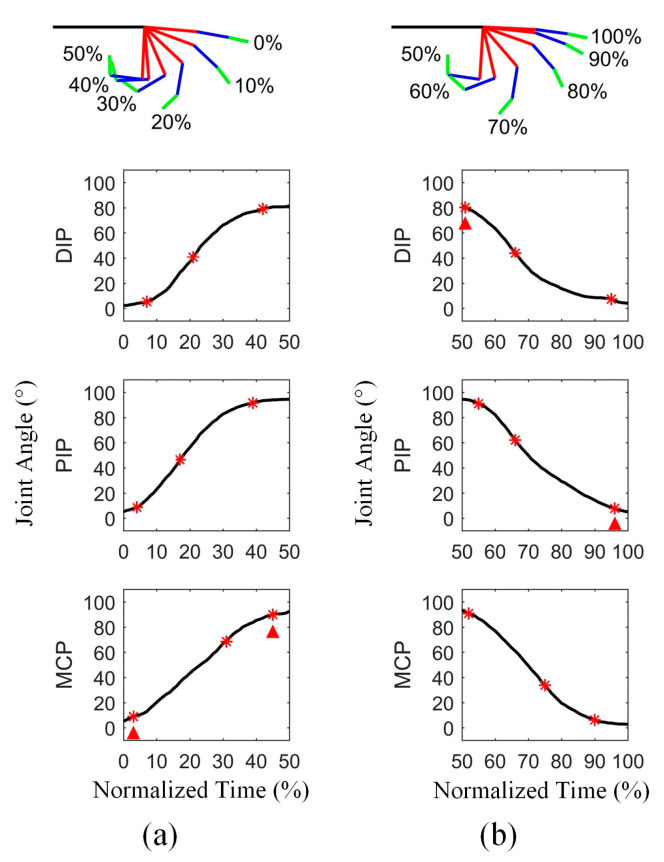
An example of finger motion and the corresponding angular curves of three joints: (**a**) flexion and (**b**) extension of middle finger from a participant. The sub-figures in the top row display finger motion, with green line for distal phalanx, blue line for middle phalanx, red line for proximal phalanx and black line for metacarpal bone. The sub-figures in the bottom three rows show the angle changes of three joints. The red asterisks represent the motion feature points of joints, including the initiation point, maximum velocity point and completion point. The red triangles represent the motion feature points of finger, including the initiation point and completion point.

**Figure 3 biomimetics-08-00244-f003:**
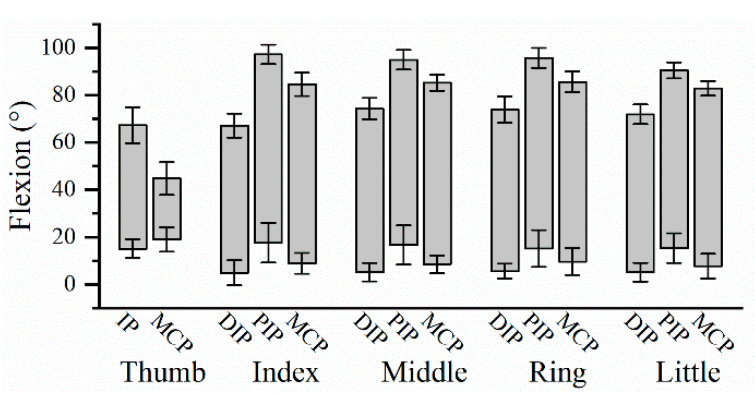
The dynamic ROM of finger joints averaged from 22 participants. Error bars are SD.

**Figure 4 biomimetics-08-00244-f004:**
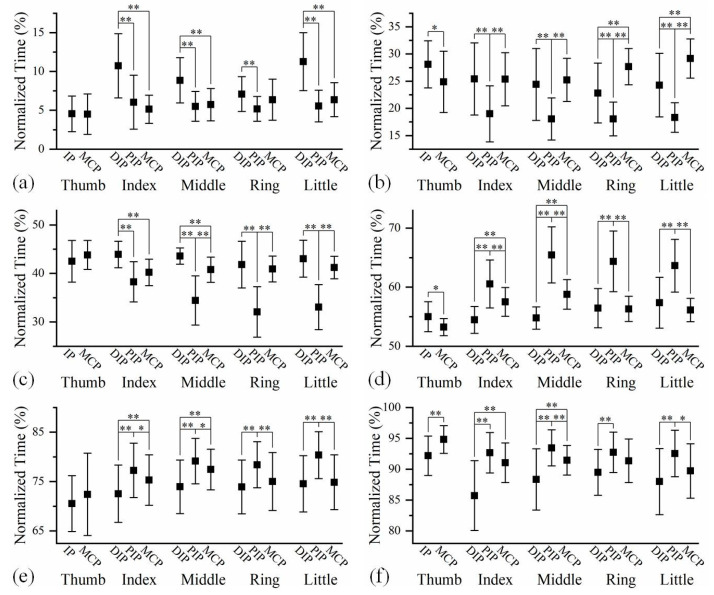
The time relationships among the feature points of joints in five fingers at (**a**) the beginning of flexion, (**b**) the middle of flexion, (**c**) the end of flexion, (**d**) the beginning of extension, (**e**) the middle of extension, and (**f**) the end of extension. The time sequences with significant differences between two joints are marked with * *p* < 0.05, ** *p* < 0.01.

**Figure 5 biomimetics-08-00244-f005:**
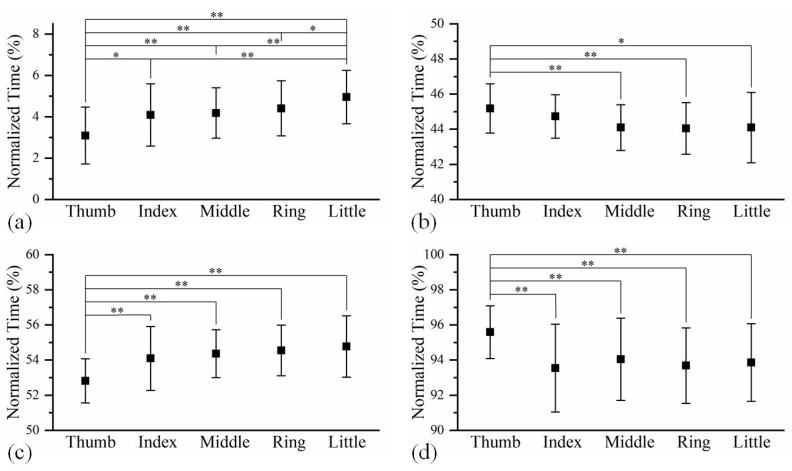
The time relationships among the five fingers at (**a**) the beginning of hand gripping, (**b**) the end of hand gripping, (**c**) the beginning of hand releasing, and (**d**) the end of hand releasing. The time sequences with significant differences between two fingers are marked with * *p* < 0.05, ** *p* < 0.01.

**Table 1 biomimetics-08-00244-t001:** The results of ANOVA on the dynamic ROM among four long fingers.

Joint	DIP	PIP	MCP
F	4.904	2.065	0.373
*p*-value	0.003	0.111	0.773

**Table 2 biomimetics-08-00244-t002:** The results of ANOVA on peak velocity among four long fingers during flexion and extension.

Joint		DIP	PIP	MCP
Peak flexion velocity	F	1.645	1.434	0.423
	*p*-value	0.185	0.239	0.737
Peak extension velocity	F	1.150	0.727	1.576
	*p*-value	0.334	0.539	0.201

**Table 3 biomimetics-08-00244-t003:** Peak velocity of thumb and finger joints during flexion and extension.

Joint	Thumb		Long Fingers		
	IP	MCP	DIP	PIP	MCP
Peak flexion velocity (°/s)	737.72 ± 196.48	374.51 ± 153.94	1110.37 ± 255.91	1393.99 ± 296.77	1260.49 ± 332.57
Peak extension velocity (°/s)	752.26 ± 191.66	335.56 ± 104.16	1144.39 ± 221.80	1368.30 ± 294.90	1269.66 ± 243.50

## Data Availability

The participants of this study did not provide consent for their data to be shared publicly.

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
