# Peer review of "Finger Kinematics during Human Hand Grip and Release"

_biomimetics, 2023, doi:10.3390/biomimetics8020244_

Round 1

Reviewer 1 Report

SUMMARY:

This work studies hand kinematic behaviour during free motion, presenting (apart from joints’ ROM) the kinematic sequence of joints and fingers, which have not been deeply studied in literature. I enjoyed with reviewing this work and I think that it provides important information to the field of human hand characterisation.

SPECIFIC COMMENTS:

TITLE

Title is appropriate and clearly reflects manuscript content.

1. INTRODUCTION

The introduction presents background enough to correctly explain the motivation of the work. Maybe it would be worthily mentioning any work studying hand kinematic synergies, as they are closely related to the kinematic sequences studied in this work.

The only comment: Please, define ROM the first time using the acronym in the main text. In the abstract it is well defined.

2. MATERIALS AND METHODS

I would suggest defining clearly the motion performed (I understand that it was free motion aiming to cover the full ROM of MCPs, PIPs and DIPs). Nevertheless, the words “grip” and “release” are confusing if no product was grasped.

3. RESULTS

Results are clear and well distributed into several sections. I would recommend specifying data shown in each box-and-whiskers plot: meaning of boxes, whiskers and IQ range, etc. Even though, plots show easily descriptive data along with ANOVA results, which ease results interpretation.

4. DISCUSSION

Discussion is well presented and results obtained are compared with those obtained in literature. Observed kinematic sequences are correlated with anatomical structure and forearm muscles, which adds a lot of consistency to results presented. This section adds a lot of strength to the study presented.

5. CONCLUSION

Conclusions are correct and highlight the main findings.

Author Response

We are grateful to your comments for the manuscript. According to your advice, we amended the relevant part in manuscript. All of your questions were answered one-by-one.

Point 1: The introduction presents background enough to correctly explain the motivation of the work. Maybe it would be worthily mentioning any work studying hand kinematic synergies, as they are closely related to the kinematic sequences studied in this work. The only comment: Please, define ROM the first time using the acronym in the main text. In the abstract it is well defined.

Response 1: “ROM” has been defined when it first appears in the main text (line 54). The entire text has been checked, and all the acronyms were defined when they first appeared in the main text.

Point 2: I would suggest defining clearly the motion performed (I understand that it was free motion aiming to cover the full ROM of MCPs, PIPs and DIPs). Nevertheless, the words “grip” and “release” are confusing if no product was grasped.

Response 2: We redescribed the motion performed in order to provide readers with a clear understanding. The description was modified to “They were asked to use right hand to perform grip and release motion as quickly as possible, while trying to ensure the fingers at maximum flexion when the hand is closed and at maximum extension when the hand is open.” (lines 93-96). Therefore, “grip” and “release” indicate closing and opening hand, respectively.

Point 3: Results are clear and well distributed into several sections. I would recommend specifying data shown in each box-and-whiskers plot: meaning of boxes, whiskers and IQ range, etc. Even though, plots show easily descriptive data along with ANOVA results, which ease results interpretation.

Response 3: Due to the limited size of the figures in the main text, we presented the detailed data corresponding to Figure 3-5 in the format of table in Supplementary Materials (form Table S1 to Table S12). Similarly, ANOVA results were presented in tables in the main text (Table 1 (line 164) and Table 2 (line 174) in main text).

Point 4: Discussion is well presented and results obtained are compared with those obtained in literature. Observed kinematic sequences are correlated with anatomical structure and forearm muscles, which adds a lot of consistency to results presented. This section adds a lot of strength to the study presented.

Response 4: Thank you for the comment.

Point 5: Conclusions are correct and highlight the main findings.

Response 5: Thank you for the comment.

Reviewer 2 Report

This paper shows finger kinematics during human hand grip and release.  The authors analyzed the motion of human finger during grip and release. The research seem to have contribution to develop a robotic hand. Also, the paper is well written and organized. The paper is worth accepting.

English looks fine.

Author Response

We are grateful to your comments for the manuscript. All of your comments were answered one-by-one.

Point 1: This paper shows finger kinematics during human hand grip and release.  The authors analyzed the motion of human finger during grip and release. The research seem to have contribution to develop a robotic hand. Also, the paper is well written and organized. The paper is worth accepting.

Response 1: Thank you for the affirmation to our study.

Reviewer 3 Report

1.  General Description 

Journal Title: Finger kinematics during human hand grip and release

Corresponding Author: Xiaodong Li

Problem Statement: To investigate finger kinematics and motion patterns of human hand to improve the performance and design of robotic hand and thus contribute to its development. The study aims to evaluate the kinematics of hand grip and release in healthy individuals by analyzing dynamic range of motion (ROM), peak velocity, joint sequence, and finger sequence.

Target audience: Engineers developing bionic robotic hands.  

Task: Investigation of human finger kinematics. 

Device: Elastic sensory glove with 15 sensors to record hand motions. 

2.  Major Comments 

The most important points identified when reviewing the journal are as follows:

·       Ambiguities and questions raised by reviewing the body of the journal: 

o   Section 2.3 Measures: It is good practice to show sample calculations or to provide a more detailed explanation as to how the variables were calculated (lines 97-98). With regards to time normalization of data, providing sample raw input data is desirable to demonstrate to the reader how input processing took place.  

o   Section 4 & 5 Discussion & Conclusion: Provide more detail as to why only the dominant hand of the healthy subjects were chosen for the investigation. Further, discuss the trade-offs between conducting the experiment on healthy subjects instead of ones with hand impairments and disabilities. Since the primary beneficiaries of a well-designed robotic hand are subjects with impaired hand functionalities, wouldn’t the investigation be more novel if it was conducted on both patients and healthy subjects? The journal included minimal discussion with regards as to how the study on healthy subjects helps engineers design better robots for non-healthy subjects. The authors only touched the surface and did not provide in-depth discussion about this matter. While they claim to offer a “guidance for the development of the robotic hand” (Section 5. Conclusion, line 279), there was little evidence to support such claim. Perhaps parts of the Discussion should be revised and rewritten. 

·       The following points should be considered for future investigations: 

o   Include “Future Work” discussions. 

o   Increasing number of human subjects by including patients who suffer from minimal hand dysfunctionalities.

o   Conduct experiments for the non-dominant hand. 

o   Perform hand grip and release experiment with objects of different shape and size. 

3.  Minor Comments 

Below is a list of minor errors found in the journal: 

·       Page 3, lines 104-105: “A diagram showing finger flexion and extension...” which of the four long fingers? Figure 1 caption should be revised to indicate what the colors (green, blue, red) at the top of the figure signify (for the grip patterns). 

·       Page 4, lines 156-159: Numbers in the brackets “(DIP joints: F(3,84)=1.645, P=0.185..” can be summarized in a table and presented in the appendix or body to enhance reading experience and paper organization (if page limit is not a concern). 

·       Page 7, lines 260-261: Spelling error in “… place their thumb outside of the first during fist clenching…” Should read fist? Double-check typo.  

There are some typos in the paper. For example:

Page 7, lines 260-261: Spelling error in “… place their thumb outside of the first during fist clenching…” Should read fist? Double-check typo.  

Author Response

We are grateful to your comments for the manuscript. The reply to your comments is attached.
